# Diagnostic Utility of Selected Serum Dementia Biomarkers: Amyloid β-40, Amyloid β-42, Tau Protein, and YKL-40: A Review

**DOI:** 10.3390/jcm9113452

**Published:** 2020-10-27

**Authors:** Karolina Wilczyńska, Napoleon Waszkiewicz

**Affiliations:** Department of Psychiatry, Bialystok Medical University, 16-070 Choroszcz, Poland; napwas@wp.pl

**Keywords:** Alzheimer’s disease, vascular dementia, mixed dementia, serum, biomarkers, amyloid beta, tau protein, YKL-40

## Abstract

Introduction: Dementia is a group of disorders that causes dysfunctions in human cognitive and operating functions. Currently, it is not possible to conduct a fast, low-invasive dementia diagnostic process with the use of peripheral blood biomarkers, however, there is a great deal of research in progress covering this subject. Research on dementia biomarkers in serum validates anticipated health and economic benefits from early screening tests. Biomarkers are also essential for improving the process of developing new drugs. Methods: The result analysis, of current studies on selected biomarker concentrations (Aβ40, Aβ42, t-tau, and YKL-40) and their combination in the serum of patients with dementia and mild cognitive disorders, involved a search for papers available in Medline, PubMed, and Web of Science databases published from 2000 to 2020. Results: The results of conducted cross-sectional studies comparing Aβ40, Aβ42, and Aβ42/Aβ40 among people with cognitive disorders and a control group are incoherent. Most of the analyzed papers showed an increase in t-tau concentration in diagnosed Alzheimer’s disease (AD) patients’ serum, whereas results of mild cognitive impairment (MCI) groups did not differ from the control groups. In several papers on the concentration of YKL-40 and t-tau/Aβ42 ratio, the results were promising. To date, several studies have only covered the field of biomarker concentrations in dementia disorders other than AD. Conclusions: Insufficient amyloid marker test repeatability may result either from imperfection of the used laboratorial techniques or inadequate selection of control groups with their comorbidities. On the basis of current knowledge, t-tau, t-tau/Aβ42, and YKL-40 seem to be promising candidates as biomarkers of cognitive disorders in serum. YKL-40 seems to be a more useful biomarker in early MCI diagnostics, whereas t-tau can be used as a marker of progress of prodromal states in mild AD. Due to the insignificant number of studies conducted to date among patients with dementia disorders other than AD, it is not possible to make a sound assessment of their usefulness in dementia differential diagnostics.

## 1. Introduction

### 1.1. Dementia

Dementia is a group of multiple etiology cognitive disorders that result in daily life impediments. The most common cause of dementia is Alzheimer’s disease (AD) which is chronic, progressive, and leads to death of the neurodegenerative process [1].

There are two main groups of pathologies that cause dementia, i.e., neurodegenerative diseases and secondary dementias [1]. The cause of neurodegenerative diseases (proteinopathies) is accumulation and aggregation of proteins with abnormal conformation [2], which results in death of neurons and supporting cells, leading to cognitive and motor dysfunctions [3]. Death of cells results in activation of glial cells and secretion of cytokine and chemokine by them, which leads to the chronic inflammatory process interfering with brain tissue homeostasis. Moreover, neurotoxicity of some inflammatory response products intensifies the neurodegenerative process [4]. Neurodegenerative dementia includes the following: dementia in Alzheimer’s disease, dementia with Lewy bodies (DLB), frontotemporal dementia (FTD), dementia in Parkinson’s disease (PDD), Huntington’s disease (HD), and dementia in prion diseases. 

The most common secondary dementia is vascular dementia. Approximately 10–15% of patients are only diagnosed with vascular-only dementia (VD). VD consists of subtypes of different etiology and clinical features, such as multi-infarct dementia, small vessel disease, strategic infarct dementia, hypoperfusion dementia, haemorrhagic dementia, hereditary cerebral autosomal dominant arteriopathy with subcortical infarction, and leukoencephalopathy (CADASIL) [1,5]. 

The observed symptoms of dementia such as depression, delirium, thyroid hormone abnormalities, vitamin deficiencies, or normotensive hydrocephalus are in potentially reversible states [1]. Chronic alcohol use is also a very common cause of dementia. In contrast to Wernicke–Korsakoff syndrome associated primarily with memory impairment, in alcoholic dementia, there are disorders in many other cognitive functions. Alcoholic dementia constitutes approximately 10% of early-onset cases and slightly more than 1% of late-onset dementia cases [6].

Dementia can be of mixed etiology, in up to half of patients, especially in senior age groups [1]. Vascular lesions occur in 40% of demented patients and usually coexist with other dementia causes. Changes caused by microinfarcts were found in around half of the patients with AD [7]. Contrary to previous diagnostic recommendations considering cerebral vascular lesions as an exclusion criterion of AD, the American Alzheimer’s Association currently takes the view that dementia often results from both neurodegenerative and vascular pathologies and suggests classifying such conditions as mixed dementias [1]. An autopsy study by Schneider et al. revealed that subjects with multifactorial etiology of dementia found in the autopsy were more frequently diagnosed with dementia during life as compared to those with only one type of neuropathological change [8]. It suggests that mixed dementia (despite its heterogeneous etiology) may be a category for underdiagnosed cases, and it also has an important clinical and prognostic value. Other relatively common forms of mixed dementia include the coexistence of AD and DLB or PDD [8].

### 1.2. Amyloid Beta and Its Role in Dementia Process

Amyloid is a group of insoluble beta-sheet proteins that form filamentous structures. They may play a role in the pathogenesis of various diseases when accumulating as deposits in the extracellular space. Each amyloid type is formed from its specific precursor protein-like amylin, derived from the precursor islet amyloid polypeptide (IAPP), involved in the pathogenesis of insulin-dependent diabetes, or AL and AH proteins composed of light and heavy chains of immunoglobulin, resulting in primary systemic amyloidosis [9]. Except for brain tissue, the amyloid precursor protein (APP) has been identified in the thymus, heart, lungs, kidneys, muscles, adipose tissue, liver, spleen, skin, and the intestine [10]. 

Amyloid beta (Aβ) refers to oligopeptides composed of 40–42 amino acid residues, originating from the amyloid precursor protein (APP). The APP protein family includes two similar peptides, i.e., amyloid precursor-like proteins 1 and 2 (APLP1 and APLP2) [11]. The APP protein encoding gene is located on chromosome 21q21. Most of the many APP mutations exist around sequences encoding secretase splitting sites and they are responsible for early-onset, familial forms of the disease [12]. The amyloid precursor protein is transformed to oligopeptides by three enzymes, i.e., α-, β- and γ-secretase. The C-terminal fragment of APP is always cut off by γ-secretase, but the forming of Aβ is determined by enzyme activity cutting off the N-terminus of either α- or β-secretase precursor peptides. Formation of Aβ happens when an N-terminus fragment is cleaved by β-secretase, while the activity of secretase α results in making a so-called secreted form of APP (s-AAPα) [13], a signal protein which is a neuron growth factor that promotes synaptogenesis and has a positive role in memory and learning processes [10]. Another factor contributing to Aβ formation are mutations of the γ-secretase subunits, i.e., presenilins. On the one hand, presenilins of correct structure regulate intracellular calcium balance, lysosome, and proteasome activity and counteract the effects of oxidative stress. On the other hand, presenilin mutations contribute to the formation of longer Aβ chains, which are more prone to aggregate into insoluble Aβ forms [14]. 

Aβ oligopeptides vary in the number of amino acids in the molecule. Most scientific research has focused on the fragment with 42 amino acid residues, i.e., Aβ42. It is the main component of amyloid plaques found in the brain of patients with AD. A second dominant isoform of Aβ is peptide with a length of 40 amino acid residues. Aβ40 is a less amyloidogenic form. Researchers have also suggested their role in preventing formation of Aβ deposits. Calculation of the Aβ42/Aβ40 ratio can be found in many scientific papers. A high value of this ratio is used as a predictor of cerebral amyloidopathy. Other forms of Aβ were also considered to be potential biomarkers, mainly Aβ38 [15]. 

In addition to previously described overproduction, accumulation, and aggregation of Aβ in the brain tissue, the impaired pathological protein degradation also contributes to dementia pathogenesis. Peroxidation of the cell membrane lipids under oxidative stress results in 4-hydroxynonenal (HNE) secretion, which has an ability to modify protein structure and function. HNE lowers the catalytic activity of neprilysin, a metallopeptidase preventing the deposition of Aβ through degradation of mono- and oligomeric Aβ forms [16]. Apolipoproteins E2 and E3 (but not E4 which is one of the main AD risk factors) are capable of binding HNE through cysteine, lysine, and histidine residue, preventing the damage of other proteins [17]. HNE also increases the synthesis of Aβ by its binding with nicastrin, part of the gamma-secretase complex, increasing activity of this enzyme [18]. 

The role of Aβ in AD pathogenesis is described in the so-called amyloid cascade hypothesis. This theory suggests that the disease process begins with overproduction and accumulation of Aβ as amyloid plaques. Their presence leads to mitochondrial and synaptic damage, disrupting the homeostasis of brain tissue. This process is accompanied by microglia and astrocytes activation, which results in inflammation and oxidative stress, eventually causing the death of neurons. Furthermore, Aβ is considered to be a factor that activates hyperphosphorylation of tau proteins. In addition to the amyloid cascade hypothesis, there are other possible explanations of the AD pathogenesis, such as the “prion-like” action of the Aβ hypothesis [19]. More and more reports have shown that the amyloid cascade hypothesis has not fully explained the pathogenesis of AD. It has recently been demonstrated that amyloid-dependent memory and synaptic plasticity impairments could occur without tau [20].

On the basis of current knowledge, pathologies tied with Aβ are not specific for AD dementia. The existence of amyloid deposits have also been identified by PET scan in dementias other than AD, especially among seniors who have been carriers of the APOE4 allele. Ossenkoppele et al. obtained positive amyloid-PET results in as many as 83% of 80-year-old APOE4 gene carriers with DLB, 43% with FTD and 64% with VD. Among a younger group of people (around 60 years old), who were not carriers of APOE4, those percentages were respectively 29%, 5%, and 7% [21]. It is not easy to state whether those observations might result from incorrectly established clinical diagnosis or rather amyloid deposition is secondary to neuropathology of other dementia types [22]. Among non-AD dementias, the clearest relationship occurs between Aβ and VD. It is known that apart from the predominant atherosclerotic etiology, other angiopathies such as congophilic amyloid angiopathy (CAA) can initiate the disease process [5]. Contrary to AD, where amyloid deposits locate mainly (but not exclusively) into intercellular space, in CAA subjects, the lesions are predominately in the blood vessel walls [3]. It has been proven that the density of congophilic vascular lesions increased with the severity of cognitive impairment [7]. On the one hand, overproduction or impaired elimination of Aβ can also occur in other diseases. Increased amyloid peptide concentrations were found in patients diagnosed with liver tumors [23], kidney [24] and liver [25] failure, Parkinson’s disease [26], obesity and insulin resistance [27], or Down Syndrome [28]. On the other hand, a decrease in Aβ42 levels in the serum has been observed in depression [29]. An increased level of Aβ in many conditions results in serious limitations of its use as a biomarker, especially in a multimorbid old age group. Meta-analysis published by Zhang et al. indicated that the lack of proper screening for comorbidities among the control group resulted in minor differences in the amyloid marker concentrations among groups in studies [30].

### 1.3. Tau Protein and Its Role in Dementia

Tau protein is a product of the MAPT gene, located on chromosome 17. It is expressed predominantly in the central nervous system [31]. This protein is found mainly inside neurons (especially in microtubule-rich axons), but also, to a lesser extent, in glial cells (astocytes and oligodendrocytes) [32], as well as in extracellular space [33]. It locates primarily in the cytoskeletal structures of the cell, but it has also been identified in the nuclei and centrosomes [32]. The MAPT gene product undergoes alternative splicing into six different, tissue-specific tau proteins. Depending on the number of 29-amino acid inserts (0, 1 or 2), there are three isoforms, i.e., 0 N, 1 N, and 2 N. Each isoform contains three or four microglobulin binding domain repeats (3R or 4R) [32,33]. In normal conditions, the tau protein is soluble and unfolded [33]. 

The physiological functions of the tau protein include stabilization and polymerization of microtubules, regulation of axonal transport, neuron polarization, axon growth and elongation, protection of DNA and RNA integrity, formation of cytoskeleton actin filaments, regulation of synaptic plasticity (dendritic tau protein), as well as cell cycle regulation via tyrosine kinase, membrane interactions, synaptic transmission, and regulation of NMDA transmission through interactions with the Fyn protein [32,33,34]. The tau protein undergoes post-translational modifications such as phosphorylation, O-glycosylation, advanced glycation, Maillard reaction, ubiquitination, nitration, sumoylation, proline isomerization, acetylation, and truncation. The most important one of the above is the kinase-induced phosphorylation process regulating tau distribution within the cell, the transport of organelles to the somatodendritic compartment, and enabling interactions with neurotransmitters and enzymes. Under physiological conditions, tau phosphorylation occurs in response to stressors such as insulin imbalance, hunger, hypothermia, anesthesia, glucocorticoids, opiates, or alcohol [32]. 

Phosphorylation also enables the tau protein to aggregate [31]. Excessive aggregation of the tau protein leads to formation of neurofibrillary tangles and it occurs in medical conditions called tauopathies and in the physiological aging process [32]. The presence of tau protein inclusions in neurons or glia causes tauopathies-progressive diseases associated with cognitive, behavioral, and motor impairment. Hyperphosphorylated tau protein and its isoforms, such as p-tau-217, measured in the peripheral blood are also promising AD biomarker candidates [35]. 

The pathological changes caused by the hyperphosphorylated tau protein (p-tau) can be divided into the following:Resulting from the loss of its physiological properties:Axonal transport disturbance;Actin cytoskeleton abnormalities leading to increased susceptibility of the cell to oxidative stress;Disturbed structure and function of mitochondria, disrupting their metabolism and increasing susceptibility to oxidative stress, which leads to the death of the cell.Resulting from the toxic effect of the abnormal isoform:
Activation of astrocytes and microglia to secrete proinflammatory mediators;Disruption of synaptic transmission through the accumulation of pathological tau in postsynaptic spines;Disturbances in proteasome degradation and autophagy [30,32].

Formation of tau protein deposits is the key pathophysiological process in primary tauopathies, while in secondary tauopathies, its origin is from other pathologies. Primary tauopathies include the subtypes of frontotemporal dementia, atypical parkinsonian syndromes, argyrophilic grain disease, and globular glial tauopathy. Considering complex (dependent also from amyloid beta) aetiopathogenesis, AD is qualified as a secondary tauopathy [36]. Other conditions, secondary to the tau protein pathology are the aging process (primary tauopathy related to age, astriogliopathy related to age), Down syndrome, prion diseases, post-traumatic encephalopathy (dementia pugilistica), amyotrophic lateral sclerosis, parkinsonism-dementia complex, postencephalitic parkinsonism, and some rare diseases such as progressive subcortical gliosis, diffuse neurofibrillary tangles with calcification, “tangle-only dementia”, Hallervorden–Spatz disease, Niemann–Pick type C disease, subacute sclerosing panencephalitis, myotonic dystrophy, non-guanamian motor neuron disease with neurofibrillary tangles, meningioangiomatosis, and tuberous sclerosis [32]. Neurofibrillary changes have also been identified post mortem in subjects without any dementia characteristics. There is an assumption that in such cases the disease was present in an asymptomatic state [7]. 

It is believed that the concentration of tau protein in CSF relates to the degree of brain cells damage [37]. Tau protein is secreted from neurons to the cerebrospinal fluid, where it penetrates through the blood-brain barrier and the arachnoid granules to the bloodstream, which is why it can be identified in peripheral blood [38]. Results of studies show the correlation between total tau protein (t-tau) concentration in serum and its brain tissue levels assessed by PET scan [39,40].

### 1.4. YKL-40: An Inflammatory Marker in Dementia 

More and more data show the key role of the inflammatory process (neuroinflammation) in AD pathogenesis. Most of all, microglia cells and astrocytes are present in the CNS inflammatory response. These cells are activated by proinflammatory cytokines, as well as Aβ and APP, causing the release of neurotoxic proinflammatory cytokines and reactive oxygen forms, and resulting in intensification of the inflammatory process and oxidative stress [41]. This knowledge resulted in taking the interest of inflammatory mediators as potential AD biomarkers.

Chitinase 3-like protein 1 (CHI3L1), also named YKL-40, human cartilage glycoprotein-39 (hcgp-39), or breast regression protein (BRP-39), is ranked among the chitinase family (glycosidic hydrolases). It is an acute-phase protein secreted into extracellular matrix through connective tissue cells (neutrophils, monocytes, macrophages, coming from monocytes, dendritic cells, osteoclasts, chondrocytes, synovial cells), vascular smooth muscle cells, glandular epithelium, and also thru other cells within an inflammatory state in response to inflammatory cytokines such as TNF-alpha, INF-gamma, IL-1beta, and IL-6 [42,43]. Some cancerous cells also have the ability of YKL-40 secretion [44].

In the inflammatory process, YKL-40 acts as an acute-phase protein that regulate proliferation, adhesion, migration, and cells differentiation. To date, the known functions of YKL-40 are the following:connective tissue repair process, i.e., connective tissue growth stimulation, bonding and fibrillogenesis of collagen, modulating of inflammatory cytokines impact on fibroblasts;stimulation of epithelial cells migration;modulation, adhesion, and migration of vascular smooth muscle cells;stimulation of alveoli macrophages to metalloproteinases and chemokines secretion;increase in auxiliary lymphocytes Th2 response caused by antigens;regulation of oxidative stress response;regulation of apoptosis process (i.e., prevention of epithelial cells apoptosis);stimulation of M2 macrophages differentiation;suppression of mammary gland epithelial cells differentiation [42,43,45].

Studies have described the relationship between YKL-40 and AD (occurrence of increased YKL-40 expression areas, concentrating mainly on astrocytes surface, around amyloid plaques and blood vessels, with amyloid angiopathy in patients with AD brain [46]), and the relationship between APOE4 genotype and YKL-40 concentration in cerebrospinal fluid (CSF) [47]. This was also supported by an experiment conducted by Choi et al., who found, in mice disease model, decreased APP expression and an improvement of rodents’ cognitive functions after administration of substance suppressing CHI3L1 activity [48]. The relationship between neuroinflammation and described biomarkers in AD are shown on Figure 1.

YKL-40 diagnostic utility is limited by its non-specificity. An increased concentration of this protein was found in various types of cancer, inflammatory and autoimmune diseases [43], bacterial, viral, and parasitic infections [44]. An increase in serum YKL-40 concentration was also documented in VD and mixed dementia with vascular risk factors, such as metabolic syndrome [49], coronary heart disease [50], atrial fibrillation [51], and obstructive sleep apnea [52]. Xu et al. concluded, in a group of patients with prehypertensive state, that the disease could be preceded by an increase in YKL-40 concentration [53]. In the first days after an ischaemic stroke, an increase in this marker was proportional to the volume of the stroke site and the severity of symptoms [54].

An increase in YKL-40 expression was observed in tissues and bodily fluids other than serum, for example, in motor cortex and spinal cord of patients with atrophic lateral sclerosis [55], in hippocampus of schizophrenia patients [56], or in CSF of elderly woman with suicidal thoughts [57]. On the one hand, increased YKL-40 immunoreactivity in brain, correlating with tau protein, was observed in tauopathies other than AD [58]. On the other hand, studies conducted by Isgren et al. have shown that decreased concentration of YKL-40 in CSF of patients with bipolar disorders was preceded by episodes of mania or hypomania [59]. 

### 1.5. Practical Aspects of Using Dementia Biomarkers

Many studies conducted, to date, have proven that brain lesions precede the occurrence of clinical symptoms of disease even by 20 years [1], which gives hope to use these observations for development of early detection and prevention methods. Efforts that have been undertaken to use proteins involved in disease pathogenesis as biomarkers have proven that they can inform about disease existence even in the asymptomatic stage and they also correlate with disease intensification when neuropathological changes buildup [60].

Considering potential application, the following can be singled out:predictive biomarkers for estimating disease in the preclinical stage, and for estimating clinical prognosis;diagnostic biomarkers in precise differential diagnosis;biomarkers for healing response and assessing the effectiveness of therapy;surrogate markers for estimating the influence of therapeutic intervention on selected pathophysiological processes;trait markers, strictly tied to disease characteristics (e.g., mutations);condition markers (e.g., enzymes) for monitoring progression of disease [22].

Diagnosis and monitoring of dementia are carried out mainly by clinical assessment and neuropsychological tests. It is only possible to diagnose the disease in the symptomatic stage, and the given diagnosis is considered to be probable [22]. Regular neuropsychological assessment, which is recommended as an early dementia diagnosis tool, is not fulfilling its role. The main causes of this situation are (among others) the long waiting time for an appointment and very short duration of visits. In standard healthcare, a common problem is also inaccurate screening performance and interpretation of cognitive functioning of patients. This results from insufficient training and experience of primary care staff. The key step is the referral of a patient to a specialist by a general medical practitioner. Adequate biochemical markers could be an inestimable tool in the hands of family physicians, facilitating selection of patients in need of specialist treatment [61]. Regardless of health benefits, early AD diagnosis could significantly reduce treatment and care costs generated by patients [1].

Usage of biomarkers in dementia diagnostics could aid early diagnosis, monitoring of disease severity, and recovery prognosis. It could also help in the development of more personalized pharmacotherapy by adequate recruiting and grouping of subjects, as well as assessment, in clinical trials, of test drug effectiveness and dosage [15]. The role of biomarkers in the development process of new drugs is crucial because of verification of patients included in clinical trials, for example, the PET amyloid test showed that many of the patients were falsely classified as AD patients [62]. In the future, proper selection and use of biomarkers may enable the development of personalized therapy for every individual patient, such as in modern oncological treatment standards [15,61]. However, the current goal, in studies on AD biomarkers, is to achieve a cost-efficient screening test with high negative (not necessarily high positive) predictive value, which should have the ability to identify AD cases on a large scale [61].

An ideal diagnostic marker should be characterized by at least moderate sensitivity and high specificity (>85% [22]), ease of obtaining research material, easy determination method, repeatability, and low cost [60]. It should also have the ability to identify disease in the early stage and differentiate it from other dementias, reflecting the neuropathology of the examined disease. Another important characteristic of a valuable biomarker is the absence of concentration changes due to the influence of the used symptomatic treatment [22]. Currently, on the one hand, biomarker identification in cerebrospinal fluid (Aβ42, t-tau, p-tau) requires qualified personnel that can use a more invasive, expensive, and often non-refundable procedure for material extraction [60,63]. Use of the CSF extraction procedure is limited to a wide-range population and its repeatability, which calls in to question its usability in standard healthcare or in clinical trials [15]. Survival assessment of the occurrence of Aβ deposits with the PET method is very costly and often non-refundable, moreover it is associated with exposure to a radioactive substance [63]. On the other hand, collecting a blood sample and its analysis is a routine procedure that does not require additional personnel training and it is repeatable and possible to perform in various conditions [61].

Imprecise measurement of the concentration of marker in peripheral blood due to its low blood-brain barrier (BBB) permeability [60], possible proteolysis of tested protein in peripheral blood, and its fast elimination by liver or kidneys are the main elements that limit the capability of using biomarkers from peripheral blood. Moreover, blood is a fluid with a high concentration of various proteins, posing a risk of obtaining false-positive results with insufficient specificity of applied tests [61,64]. Therefore, imperfections of available markers may be minimalized by application of several different test panels [60].

The aim of this review is to gather and summarize studies conducted to date, on using selected proteins that can act as dementia biomarkers in blood material, for example, Aβ40 and Aβ42 amyloids, tau protein, as well as its phosphorylation products (p-tau) and YKL-40 protein.

## 2. Experimental Section

For the analysis of acquired studies and their results on concentrations of selected biomarkers in the blood of dementia patients, the search for works about human research published in English scientific journals in Medline, Pubmed, and Web of Science databases was conducted with the use of keywords such as “amyloid beta”, ”tau protein”, ”YKL-40 OR CHI3L1” AND ”dementia OR alzheimer*” AND ”plasma OR serum or blood”. Considering the possibility of outdated laboratory techniques influencing results, the search was limited only to studies from 2000 to 2020. Among the acquired search results, only cross-sectional studies with a control group and a test group with dementia and mild cognitive impairments (MCI) were qualified for comparison. Review articles and prospective studies were excluded. Studies on biomarkers in other than plasma or serum biological material, as well as studies on specific isoforms of the markers, were excluded. We also did not qualify research made on specific groups, i.e., corpses, children, patients with concomitant diseases (e.g., Down syndrome), or patients with undefined dementia etiology. Some groups of patients were investigated in more than one article, in such cases, only one methodologically best study was taken into account. Papers with no significance level “*p*” value for comparison among groups and in which test or control groups consisted of less than 10 people were also excluded. The results obtained in groups other than dementia and MCI were not included in the tables. The number of search results in each database is listed in the Table 1.

## 3. Results

### 3.1. Amyloid Markers Aβ40, Aβ42, and Aβ40/42 Ratio

According to the adopted criteria, 50 studies comparing concentrations of Aβ40, Aβ42, and their ratio were selected from the search results. A total number of 7303 patients with dementia or MCI and people from control groups without any cognitive disorders were examined in these studies. In 31 studies, the ELISA method was used for identification; in eight studies, the immunomagnetic reduction method (IMR) was used; in six studies, the fluorescence in multiplex immunology test method (xMAP) was used; and in two studies, the single molecular array method (Simoa) was used. In one study, identification was made using the immunoblot method [65]. The most recent from the covered studies [66] was conducted by carbon nanotubes array (CNT). 

The Aβ40 concentration was tested in 43 trials, listed in Table 2. The obtained results are highly incoherent. Statistically significant differences among groups were found in only 24 trials, whereas, in 10 of the trails, the concentrations of Aβ40 were (contrary to general trend) higher in the control group. In 39 trials, AD patients were investigated. Twenty trials compared MCI subjects to cognitively normal controls. The inconsistency of results did not depend on the method used. A large part of research gave statistically not significant results (12 out of 20 for ELISA, three out of eight for IMR, two out of five for xMAP). 

In 48 of the selected studies, there was a comparison of Aβ42 among groups. In 32 of them, the results were statistically significant. In most of the comparisons, the Aβ42 concentrations were higher in control groups, but in as many as 13 cases higher concentrations of Aβ42 were identified in patients with dementia or MCI rather than in the control group. Among the methods repeated in several studies, it was noteworthy that, in all seven studies conducted using the IMR method, the Aβ42 levels were significantly higher in the AD or MCI groups than in the controls. In addition, studies based on ELISA or xMAP method gave very inconsistent results. 

The amyloid peptide concentration ratio was assessed in 29 studies. In nine cases, the results were not statistically significant. Among the comparisons that were statistically significant, in 11 cases, the Aβ42/Aβ40 concentration ratio was higher in the control groups and, in seven cases, it was higher in the study groups. In two of the studies, the Aβ42/Aβ40 ratio was similar in the AD and control groups and significantly higher than in the MCI groups [68,102]. The results varied depending on laboratory methods use. None of the three studies using xMAP gave statistically significant results, while, in four out of five IMR studies, the Aβ42/Aβ40 ratio was significantly higher in the AD or MCI groups than in the controls. The results obtained in ELISA studies were highly inconsistent. 

### 3.2. Tau Protein

Twenty cross-sectional tests, conducted to date, on t-tau concentration in patients with dementias and MCI serum, and which meet the search criteria, are listed in Table 3. One work included two separately examined cohorts [114]; 16 patients with AD and 12 patients with MCI were examined. Single tests referred to FTD (t-tau significantly higher than in control group [115]) and VD (t-tau significantly lower than in AD and higher than control group [87,109]). The identification of tau protein was conducted in seven studies by the Simoa method, in six studies by the ELISA method, in six studies by the IMR method, and in one study by CNT. The absolute values of t-tau concentration identified by the single molecule array (Simoa) method were lower by one to two orders of magnitude than values acquired by other methods.

In 14 studies, t-tau concentration was higher as compared with a control group; in five studies, there was no significant difference; and in two studies [87,95], t-tau concentration was higher in the control group. A significantly higher concentration of t-tau protein in AD as compared with a control group was identified in 13 out of 16 studies, and AD as comparing with MCI in eight of nine studies. In addition, in studies comparing t-tau concentration between MCI and the control group, results indicating diagnostic usability of this marker were achieved in only two out of 12 comparisons. 

In all six studies using the IMR method and in five out of six studies performed with ELISA determination, tau levels were significantly higher in AD/MCI patients. Simoa gave less consistent results, i.e., in three out of eight cohorts the differences among groups did not reach the statistical significance.

### 3.3. Amyloid Markers and Total Tau Protein Combinations

In database searches, there were four studies on t-tau/Aβ42 ratio as a dementia biomarker in serum. The results are listed in Table 4. In three studies, the t-tau/Aβ42 concentration ratio was higher in the control group than in the AD group, but, in one of them, this difference was not statistically relevant. The age gap between the AD groups and control groups could have influenced the obtained results from this study [28]. In the Krishnan et al. study, the combination of both markers was characterized by significantly higher diagnostic sensitivity in differentiation of AD from healthy subjects (90.3%, AUC 0.991), than each separate marker (80.6%), but specificity for t-tau, Aβ42, and markers ratio hovered around 67% [87]. Kim et al. acquired a higher t-tau to Aβ42 concentration ratio level in the test group, using a new identification technique [66]. Assessment of t-tau/Aβ42 in serum was also conducted by Park et al. Authors of this study did not provide a comparison among groups (AD, MCI, HC), but they were searching for a correlation of acquired biomarkers ratio to existence or non-existence of tau protein deposits by the use of PET. The researchers managed to show a relation between biomarkers concentration ratio and occurrence of tauopathy within cingulate gyrus, temporal, prefrontal, and orbitofrontal cortex [40].

Moreover, diagnostic usability of the t-tau/Aβ42 ratio identified by IMR method was assessed in two other studies. The results obtained by Chiu et al. pointed to 80% sensitivity and 82% specificity of the biomarker ratio in AD as compared with MCI, as well as 96% sensitivity and 97% specificity for the cognitive impairments group (AD and MCI) as compared with healthy subjects. The authors did not provide the *p* value for comparisons among groups [123]. In the Tsai et al. study, the difference between a cognitively impaired group and a control group was statistically relevant (*p* < 0.05), but in the AD and MCI groups the acquired values of the biomarker ratio overlapped (respectively 473.2 ± 107.4 and 418.3 ± 80.3) [111].

### 3.4. YKL-40 

The adopted search criteria were met by five studies that focused on diagnostic and prognostic usability assessment of YKL-40 in MCI and dementia in serum of patients. Their results are listed in Table 5.

Briefly, the findings of these five studies were the following:Craig-Schapiro et al. found a statistically relevant increase in YKL-40 concentration in serum with disease progress which corresponded with results from CSF [45].Choi et al. suggested that YKL-40 could be highly useful in the diagnostic of MCI progression to mild AD. This was supported by the highest marker increase among these groups and the correlation of acquired results with patient functioning, measured by the CDR scale at this stage of disease progression diagnosis [127]. The results of this study do not clearly reinforce the prognostic value of YKL-40 in serum, but they suggest its diagnostic usability in combination with other markers (Aβ42, tau, and p-tau) [124].Grewal et al. examined women with amnestic MCI subtype from various ethnicity groups. Statistically significant concentration differences of YKL-40 among subgroups were identified only in the Latin American group. In addition, other measured biomarkers showed the differences of sensitivity among ethnic groups [93]. Amnestic type MCI is a state highly predisposing to AD, contrary to non-amnestic types which are the bases for developing other dementias. The results of prospective studies show that, every year, dementia is developed in 10–15% of patients with MCI [128].Surendranathan et al. did not shown any statistically significant YKL-40 concentration differences between patients with DLB and a control group [125].Villar-Pique et al. did not obtain any statistically relevant differences between AD patients and a control group. The acquired results were significantly divergent across groups. Among the tested groups, highly increased YKL-40 concentrations were observed in patients with CJD (*p* < 0.001) and to a lesser degree in patients with LBD (*p* < 0.05). The authors hypothesized that the crucial factor influencing the concentration of the marker in peripheral blood may be the damage level of the blood-brain barrier in the course of primary disease [126].

## 4. Discussion

In dozens of studies, conducted to date, comparing the concentrations of Aβ40, Aβ42, and Aβ42/Aβ40 in blood of dementia and MCI patients, the results have been incoherent. This constitutes a fairly surprising observation regarding the crucial role of Aβ in AD pathogenesis and proven diagnostic value of amyloid markers in CSF. The incoherence of the observed results suggest that Aβ concentration in serum does not reflect its level in brain tissue and CSF. It is equally probable that not enough restrictive selection of a control group regarding comorbidities or imprecision of used laboratorial methods could have had an influence on these results. 

Tau protein is characterized by higher specificity than other markers. In common, typically for the elderly somatic disorders, there was no increase in its concentration. The concentration of tau protein increases mainly in tauopathies, from which many of them are rare diseases. Therefore, t-tau may be the preferred marker in patients with somatic ailments. The most common tauopathy, other than AD, is FTD. The t-tau concentration in serum is increased in both of these diseases, however it seems that regarding specific FTD symptoms (behavioral problems preceding dementia and distinctive aphasia) differentiation of these diseases may not be a major diagnostic problem. According to many experts, tau protein hyperphosphorylation and its accumulation in the form of neurofiber tangles is secondary to amyloidopathy. The results of studies have highlighted that the lack of an increase in t-tau, in patients with MCI, could possibly be confirmation of this theory. T-tau may, thus, serve as a marker of progression from MCI to AD. However, it seems useless in asymptomatic stages and MCI diagnostics. 

The past failure to identify a single, sensitive, and specific dementia serum marker implies attempts using combinations of more than one biomarker. In several studies, conducted to date, comparing values of t-tau and Aβ indicator, the acquired results were promising, and therefore this seems to be a good reason for further studies.

There are not many works on using YKL-40 in dementia diagnostics. Studies by Craig-Schapiro et al. [45], Choi et al. [124], and Grewal et al. [93] have given hope for its use in diagnostics of early stage dementias (MCI and mild stage AD). Less promising results were acquired by Villar-Pique et al. [126]. Due to the increase in this marker in many other diseases, there is an assumption that its usage may be limited only to the patients without any somatic comorbidity. 

Considering the results of amyloid markers separately for individual laboratory methods, we noticed some differences. In the case of amyloid markers, on the one hand, almost all tests performed with the IMR method gave consistent, statistically significant results, in contrast to the determinations by ELISA or xMAP. On the other hand, the Simoa method, often used for the determination of tau protein concentrations, in the analyzed works, gave much less consistent results as compared with ELISA and IMR. Therefore, it seems that the laboratory method used may influence the obtained results.

## Figures and Tables

**Figure 1 jcm-09-03452-f001:**
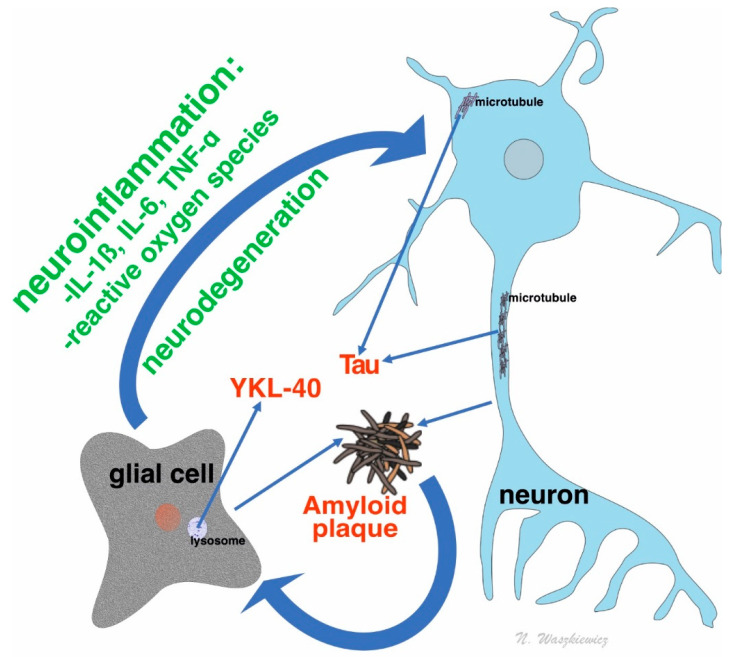
The role of neuroinflammation and neurodegeneration processes in the formation of dementia protein biomarkers, i.e., amyloid, tau, and YKL-40. IL-1b, interleukin 1 beta; IL-6, interleukin 6; TNF-a, tumor necrosis factor alpha.

**Table 1 jcm-09-03452-t001:** Search results in each database.

Biomarker	Database	Found	Qualified	Qualified Total
Amyloid beta	Medline	1698	41	50
	PubMed	1709	40	
	Web of Science	7133	48	
T-tau	Medline	405	15	20
	PubMed	397	16	
	Web of Science	1547	14	
YKL-40	Medline	14	4	5
	PubMed	11	3	
	Web of Science	38	5	
Amyloid beta and t-tau combination	Medline	281	5	4
	PubMed	255	5	
	Web of Science	960	5	

**Table 2 jcm-09-03452-t002:** Studies on Aβ40, Aβ42, and Aβ40/Aβ42 in serum.

Study	N	Groups (*n*)	Aβ40 Concentration [pg/mL] (SD or CI)	*p*	Aβ42 Concentration [pg/mL] (SD or CI)	*p*	Aβ42/Aβ40 (SD or CI)	*p*	Method
Mehta et al., 2000 [67]	65	AD (36), HC (29)	AD 272 (100–770) HC 219 (35–490)	0.005	AD 73 (25–880) HC 81 (25–905)	>0.05	-	-	ELISA
Sobów et al., 2005 [68]	158	AD (54), MCI (39), HC (35)	AD 168.7 (32.2) MCI 160.1 (20.2) HC 160.1 (15.2)	>0.05	AD 37.8 (10.3) MCI 56.8 (9.3) HC 36.3 (6.3)	<0.001 (MCI vs. AD + HC)	AD 4.6 (0.9) MCI 2.9 (0.6) HC 4.5 (0.6)	<0.001 (MCI vs. AD + HC)	ELISA
Pesaresi et al., 2006 [69]	324	AD (146), MCI (89), HC (89)	-	-	AD 38 MCI 52 HC 54	<0.01 (AD vs. MCI + HC)	-	-	ELISA
Fagan et al., 2007 [70]	114	AD CDR1 (16), AD CDR0.5 (33), HC-CDR0 (65)	AD CDR 1 AD CDR 0.5 CDR 0 191 (61.3)	0.51	AD CDR1 36 (37.2) AD CDR0.5 41 (38.9) CDR0 36 (29.4)	0.76	Aβ40/42 AD CDR1 9.25 (7.0) AD CDR0.5 7.78 (6.5) CDR0 8.64 (8.9)	0.82	ELISA
Abdullah et al., 2007 [71]	213	AD (67), HC (146)	Serum AD 183.01 (6.23) HC 134.33 (2.79) Plasma AD 103.38 (4.27) HC 80.66 ± 1.86	<0.05 <0.01	Serum AD 9.87 (0.82) HC 9.02 (0.40) Plasma AD 4.53 (0.52) HC 5.66 (0.31)	>0.05 >0.05	Serum AD 0.064 (0.011) HC 0.076 (0.005) Plasma AD 0.053 (0.008) HC 0.061 (0.006)	<0.05 >0.05	ELISA
Baranowska-Bik et al., 2008 [72]	124	mildAD (29), m-sAD (28), HC (67)	-	-	HC > mildAD > mod-sevAD	<0.05 (mild vs. m-sAD) <0.01 (HC vs. mildAD)	-	-	ELISA
Xu et al., 2008 [73]	268	AD (113), HC (155)	AD 112 (39.51) pmol/L HC 95.38 (32.30)	<0.0002	AD 10.29 (13.80) pmol/L HC 12.13 (12.29)	<0.0001	Aβ40/42 AD 14.42 (10.00) HC 8.34 (3.83)	<0.0001	ELISA
Ait-ghezala et al., 2008 [74]	175	AD (73), HC (102)	AD 91.99 (5.02) HC 81.04 (2.94)	<0.05	AD 1.91 (0.14, 6.84) HC 2.82 (0.59, 5.38)	>0.05	AD 0.015 (0.002, 0.097) HC 0.032 (0.008, 0.065)	>0.05	ELISA
Roher et al., 2009 [75]	38	AD (17, HC (21)	AD 424.06 (147.73) HC 344.41 (132.43)	0.088	AD 139.91 (77.82) HC 124.71 (42.34)	0.448	AD 0.35 (0.16) HC 0.44 (0.30)	0.292	ELISA
Sedaghat et al., 2009 [76]	35	AD (29), HC (16)	-	-	AD 16.2 (2.6) HC 13.4 (1.4)	>0.05	-	-	ELISA
Luis et al., 2009 [77]	78	AD (25), MCI (13), HC (40)	AD 181 (13.78) MCI 158 (17.55) HC 158 (7.65)	>0.05	AD 13.89 (2.00) MCI 23 (5.93 pg/mL) HC 10 (1.84)	0.015 0.02 (MCI vs. HC)	AD 0.086 (0.013) MCI 0.161 (0.045) HC 0.071 (0.014)	0.021	ELISA
Cammarata et al., 2009 [78]	293	MCI (191), HC (102)	MCI 294.7 (20.86) HC 315.6 (23.64)	>0.05	MCI 26.62 (2.68) HC 16.46 (1.46)	<0.01	MCI 0.12 (0.02) HC 0.09 (0.01)	<0.01	ELISA
Lui et al., 2010 [79]	1032	AD (186), MCI (122), HC (724)	AD 155.1 (44.2) MCI 152.9 (51.5) HC 153.4 (40.2)	0.877	AD 30.0 (10.2) MCI 30.2 (11.9) HC 32.4 (9.7)	<0.001	AD 0.199 (0.056) MCI 0.216 (0.120) HC 0.221 (0.097)	0.001	ELISA
Konno et al., 2011 [80]	49	AD (39), HC (21)	AD 378 (113) HC 254 (63)	<0.0001	-	-	-	-	ELISA
Han et al., 2012 [81]	343	AD (112), VD (85), other dementias—OD (30), HC (116)	AD 90.7 (8.7) VD 93.6 (12.3) OD 93.1 (11.0) HC 92.4 (13.0)	>0.05	AD 32.1 (3.0) VD 37.3 (7.5) OD 37.2 (5.5) HC 37.7 (7.6)	<0.001	AD 0.29 (0.07) VD 0.4 (0.09) OD 0.4 (0.08) HC 0.41 (0.09)	<0.001	ELISA
Zhang et al., 2013 [82]	326	AD (153), VD (53), HC (120)	AD 97.7 (30.6) VD 98.2 (20.5) HC 92.6 (26.7)	>0.05	AD 11.5 (2.9) VD 13.2 (3.1) HC 13.3 (3.7)	<0.001 (AD vs. HC) <0.01 (AD vs. VD)	AD 0.12 (0.03) VD 0.14 (0.02) HC 0.14 (0.01)	<0.001 (AD vs. HC, AD vs. VD)	ELISA
Huang et al., 2013 [83]	34	AD (18), MCI + HC (16)	-	-	AD 17.19 (21.9) MCI + HC 7.31 (5.3)	0.079	-	-	ELISA
Ruiz et al., 2013 [84]	140	AD (51), MCI (36), HC (53)	AD 51 (16) MCI 58.9 (16) HC 44.4 (14)	<0.002	AD 10.8 (7.5) MCI 14 (18) HC 13 (12)	<0.002 (n/s)	-	-	ELISA
Wang et al., 2014 [85]	273	AD 97 MCI 54 HC 122	AD 59.10 (20.30) MCI 51.66 (26.03) HC 43.14 (22.57)	MCI vs. HC 0.027 MCI vs. AD 0.063 AD vs. HC <0.001	AD 47.10 (2.29) MCI 47.49 (0.93) HC 47.53 (1.97)	0.944 MCI vs. HC 0.474 MCI vs. AD 0.468 AD vs. HC	-	-	ELISA
Tzikas et al., 2014 [86]	55	AD (28), HC (27)	AD 39.65 (8.08) HC 36.30 (6.68)	0.171	AD 3.38 (2.34) HC 3.39 (2.64)	0.849	-	-	ELISA
Krishnan et al., 2014 [87]	105	AD (30), VD (35), HC (40)	-	-	AD 164.66 (66.76) VD 148.17 (60.24) HC 86.10 (43.75)	<0.001 (AD vs. HC, VD vs. HC) >0.05 (AD vs. VD)	-	-	ELISA
Kleinschmidt et al., 2015 [88]	94	AD (15), MCI (14), HC 18–30 years (13), HC 40–65 (13), HC 66–85 (19)	HC 66–85 > AD > HC 40–65 > MCI > HC 18–30	<0.05 for AD vs. MCI <0.01 for MCI vs. HC 66–85	HC 66–85 > HC 18-30 > HC 40–65 > AD > MCI	<0.05 for AD vs. HC 66-85 and MCI vs. HC 66-85	HC 18-30 > HC 40–65 > HC 66–85 > MCI > AD	<0.01 (AD vs. HC 66–85) <0.05 (MCI vs. HC 66–85)	ELISA
Jiao et al., 2015 [89]	285	AD (156), HC (129)	AD 86.2 (55.5) HC 60.2 (34.7)	<0.001	AD 68.4 (61.9) HC 49.3 (27.7)	0.001	-	-	ELISA
Igarashi et al., 2015 [90]	153	AD (70), MCI (50), HC (33)	Median AD 51.0 pmol/L MCI 50.0 HC 51.4	>0.05	Median AD 6.4 pmol/L MCI 6.2 HC 6.9	>0.05	Median Aβ40/42 AD 8.2 MCI 7.8 HC 6.9	0.01 (AD vs. HC) <0.05 (MCI vs. HC)	ELISA
Kim et al., 2015 [91]	146	AD (100), HC (46)	AD 58.7 (20.2) HC 54.2(25.0)	0.371	AD 9.0 (4.0) HC 10.4 (3.5)	0.003	Aβ40/42 AD 6.8 (2.1) HC 5.0 (1.7)	0.000	ELISA
Poljak et al., 2016 [92]	251	AD (39), MCI (93), HC (129)	AD 155.82 (75.11) MCI 233.64 (100.56) HC 254.85 (145.72)	AD vs. HC *p* < 0.001 MCI vs. HC *p* = 0.14	AD 18.34 (32.10) MCI 37.58 (74.38) HC 65.63 (217.04)	<0.001 (AD vs. HC) 0.005 (MCI vs. HC)	AD 0.20 (0.64) MCI 0.23 (0.49) HC 0.26 (0.59)	<0.001 (AD vs. HC) 0.019 (MCI vs. HC)	ELISA
Grewal et al., 2016 [93]	75	3 groups of 15 (aMCI) and 10 (HC) women of different races	LA aMCI 127.63 (23.76) CA aMCI 160.51 (25.91) AA aMCI 106.28 (9.57) LA HC 104.81 (18.66) CA HC 96.02 (20.24) AA HC 103.33 (14.77)	LA *p* < 0.05 CA *p* = 0.0001 all groups *p* = 0.0001	LA aMCI 40.38 (4.76) CA aMCI 33.21 (2.81) AA aMCI 26.48 (2.61) LA HC 23.69 (2.34) C HC 34.82 (4.00) AA HC 26.95 (4.05)	<0.005 (LA) >0.05 (CA, AA)	LA aMCI 0.3 (0.05) C aMCI 0.16 (0.01) AA aMCI 0.41 (0.19) LA HC 0.4 (0.31) C HC 0.46 (0.09) AA HC 0.3 (0.05)	>0.05	ELISA
Yamashita et al., 2016 [94]	36	AD (18), HC (18)	AD 103.6 (11.8) fmol/mL HC 81.2 (9.8)	>0.05	AD 25.0 (5.3) HC 18.5 (2.8)	>0.05	AD 0.3 (0.1) HC 0.3 (0.0)	>0.05	ELISA
Rani et al., 2017 [95]	90	AD (45), HC (45)	-	-	AD 174.87 (62.15) HC 90.62 (42.35)	<0.001	-	-	ELISA
Sun et al., 2018 [96]	137	AD (76), HC (61)	AD 215.25 (54.26) HC 144.62 (47.20)	<0.001	AD 123.48 (45.89) HC 91.35 (36.39)	<0.001	-	-	ELISA
Chen et al., 2018 [97]	126	AD (96), HC (30)	AD 649.68 (132.21) HC 423.52 (100.99)	<0.001	AD 322.25 (76.04) HC 219.21 (62.51)	<0.001	Aβ40/42 AD 2.13 (0.66) HC 2.15 (0.95)	>0.05	ELISA
Bibl et al., 2007 [65]	85	AD (15), AD-CVD (20), VD (15), PD/PDD (20), HC (15)	AD 0.199 (0.099) AD-CVD 0.197 (0.083) VD 0.270 (0.103) PD/PDD 0.185 (0.069) HC 0.209 (0.087)	<0.05 (VD vs. HC) remaining *p* > 0.05	AD 0.022 (0.007) AD-CVD 0.023 (0.013) VD 0.022 (0.008) PD/PDD 0.023 (0.007) HC 0.025 (0.007)	>0.05	-	-	immunoblot
Le Bastard et al., 2009 [98]	162	AD (48), non-AD (46), MCI (39), HC (29)	-	-	AD 40.5 (32.8–50.9) non-AD 42.1 (33.1–48.6) MCI 44.3 (38.3–55.8) HC 38.9 (31.0–46.1)	0.174	-	-	xMAP
Le Bastard et al., 2010 [99]	147	AD (50), non-AD (50), HC (47)	AD 306.8 (268.7–336.8) non-AD 292.7 (238.9–334.9) HC 284.8 (240.6–333.6)	0.347	AD 40.4 (32.1–50.8) non-AD 41.7 (33.4–48.0) HC 39.4 (29.4–46.7)	0.506	AD 0.135 (0.110–0.160) non-AD 0.152 (0.122–0.185) HC 0.137 (0.111–0.153)	0.056	xMAP
Sundelöf et al., 2010 [100]	213	AD (101), MCI (84), HC (28)	AD 145.9 (64.3) MCI 166.8 (57.1) HC 91.9 (28.5)	<0.05 (AD vs. HC and MCI vs. HC)	AD 28.5 (10.7 MCI 36.9 (11.7) HC 22.0 (9.2)	<0.05 (AD vs. HC and MCI vs. HC)	-	-	xMAP
Chou et al., 2016 [101]	781	AD (592), MCI (119), HC (170)	AD 173.1 (79.3) MCI 178.7 (54.6) HC 171.6 (64.3)	0.807 (AD vs. MCI) 0.318 (AD vs. HC)	AD 23.8 (15.1) MCI 23.6 (12.5) HC 23.7 (12.6)	0.899 (AD vs. MCI) 0.969 (MCI vs. HC)	AD 0.15 (0.25) MCI 0.14 (0.07) HC 0.15 (0.08)	0.904 (AD vs. HC) 0.189 (MCI vs. HC)	xMAP
Hsu et al., 2017 [102]	335	AD (177), MCI (60), HC (108)	AD 170.3 (63.9) MCI 171.1 (54.5) HC 143.7 (34.9)	0.0001 (AD vs. HC) 0.0013 (MCI vs. HC)	AD 37.2 (14.1) MCI 34.9 (9.5) HC 33.6 (10.2)	0.025 (AD vs. HC) 0.38 (MCI vs. HC)	AD 0.232 (0.095) MCI 0.210 (0.06) HC 0.239 (0.064)	0.14 (AD vs. HC) 0.0032 (MCI vs. HC)	xMAP
Hanon et al., 2018 [103]	1040	AD (501), aMCI (417), naMCI (122)	AD 263 (80) aMCI 269 (68) naMCI 272 (52)	0.04	AD 36.9 (11.7) aMCI 38.2 (11.9) naMCI 39.7 (10.5)	0.01	-	-	xMAP
Uslu et al., 2012 [104]	60	AD (18), MCI (16), HC (26)	AD 53.21 (34.69) MCI 47.98 (16.20) HC 65.84 (13.47)	>0.05	AD 34.22 (31.62) MCI 22.66 (20.83) HC 15.79 (0.56)	0.001 (AD vs. HC)	AD 0.6906 (0.3363) MCI 0.4502 (0.1864) HC 0.2464 (0.0370)	<0.001 (AD vs. HC and MCI vs. HC)	IMR
Chiu et al., 2012 [105]	60	AD (18), MCI (16), HC (26)	AD 53.21 (34.69) MCI 47.98 ± 16.20 HC 65.84 (13.47)	>0.05	AD 34.22 (31.62) MCI 22.66 (20.83) HC 15.79 (0.56)	0.001 (AD vs. HC)	AD 0.6906 (0.3363) MCI 0.4502 (0.1864) HC 0.2464 (0.0370)	AD vs. MCI *p* < 0.001 MCI vs. HC *p* < 0.0001	IMR
Tzen et al., 2014 [106]	45	AD (14), MCI (11), HC (20)	AD 36.9 (1.6) MCI 41.4 (1.8) HC 60.9 (6.4)	<0.001	AD 18.9 (0.3) MCI 17.2 (0.3) HC 15.9 (0.3)	<0.001	AD 0.52 (0.07) MCI 0.42 (0.07) HC 0.26 (0.03)	<0.001	IMR
Lee et al., 2017 [107]	140	AD (62), HC (78)	AD 43.9 (22.1) HC 61.1 (6.3) and 60.7 (6.9)	<0.001	AD 23.2 (18.4) HC 15.8 (0.3) and 16.0 (0.5)	<0.001	AD 0.55 (0.23) HC 0.26 (0.03) and 0.27 (0.04)	<0.001	IMR
Teunissen et al., 2018 [108]	106	AD 63 HC 43	AD 17.9 (4.3) HC 15.5 (2.1)	<0.001	-	-	-	-	IMR
Tang et al., 2018 [109]	79	AD (21), VD (34), HC (24)	HC >VD >AD	0.01 (AD vs. HC) <0.01 (VD vs. HC)	AD > VD > HC	<0.05 (AD vs. HC) <0.01 (AD vs. VD) <0.05 (VD vs. HC)	-	-	IMR
Fan et al., 2018 [110]	80	AD (16), MCI (25), HC (39)	AD 39.5 (5.8) MCI 41.5 (3.9) HC 59.2 (11.1)	<0.001 (AD vs. HC, MCI vs. HC)	AD 19.0 (2.7) MCI 17.0 (2.0) HC 16.1 (1.8)	<0.001 (AD vs. MCI, MCI vs. HC)	-	-	IMR
Tsai et al., 2019 [111]	90	AD (37), MCI (40), HC (13)	AD 51.7 (3.7) MCI 51.9 (4.9) HC 51.8 (5.1)	>0.05	AD 17.4 (1.0) MCI 17.0 (0.7) HC 16.7 (0.7)	<0.05 (AD + MCI vs. HC)	AD 0.338 (0.032) MCI 0.330 (0.035) HC 0.326 (0.035)	>0.05	IMR
Startin et al., 2019 [28]	54	AD (27), HC (27)	AD 160.80 (43.60–420.00) HC 144.40 (26.88–355.60)	0.506	AD 13.32 (4.28–18.84) HC 14.76 (2.00–45.62)	0.710	AD 0.08 (0.04–0.11) HC 0.10 (0.07–0.17)	<0.001	Simoa
Janelidze et al., 2016 [112]		AD (57), MCI (214), HC (274)	AD 244.3 (105.8) MCI 287.6 (77.0) HC 276.7 (66.1)	<0.001 (AD vs. HC) <0.0001 (AD vs. MCI)	AD 13.2 (7.3) MCI 18.8 (6.1) HC 19.6 (5.2)	<0.0001 (AD vs. HC) <0.0001 (AD vs. MCI)	AD 0.057 (0.022) MCI 0.066 (0.015) HC 0.073 (0.023)	0.0001 (AD vs. HC) 0.002 (MCI vs. HC) 0.003 (AD vs. MCI)	Simoa
Shi et al., 2009 [113]	155	MCI (68), HC (87)	MCI 157.65 (64.50) HC 183.76 (61.87)	0.011	MCI 5.95 (2.60) HC 8.14 (3.12)	0.000	-	-	Simoa
Kim et al., 2020 [66]	40	AD (20), HC (20)	AD 184 (67.8) HC 159 (78.0)	0.26	AD 6.49 (5.02) HC 19.3 (15.5)	<0.001	AD median approx. 0.1 HC median approx. 0.05 (from the graph)	<0.000001	CNT

SD, standard deviation; CI, confidence interval; AD, Alzheimer’s disease; MCI, mild cognitive impairments; HC, control group; CDR, clinical dementia rating; VD, vascular dementia; PD, Parkinson’s disease; mildAD, AD in mild level; m-sAD, moderate and severe AD; non-AD, dementia of etiology other than AD; aMCI, amnestic MCI subtype; naMCI, non-amnestic MCI subtype; ELISA, immunoenzymatic method; IMR, immunomagnetic reduction method; CNT, carbon nanotube array.

**Table 3 jcm-09-03452-t003:** Studies on t-tau concentration in serum.

Study	N	Groups (*n*)	Tau Concentration in Serum (SD or CI) [pg/mL]	*p*	Method
Chiu et al., 2014 [116]	60	AD (10), MCI (20), HC (30)	AD 53.9 (11.7) MCI 32.7 (5.8) HC 15.6 (6.9)	<0.01 (MCI vs. AD) >0.05 (MCI vs. HC)	IMR
Tzen et al., 2014 [106]	45	AD (14), MCI (11), HC (20)	AD 46.7 (2.0) MCI 33.5 (2.2) HC 13.5 (5.5)	<0.001	IMR
Lee et al., 2017 [107]	140	AD (62), HC (78)	AD 47.5 (18.9) HC 15.0 (7.3) and 14.9 (5.5)	<0.001	IMR
Tang et al., 2018 [109]	79	AD (21), VD (34), HC (24)	AD > VD > HC	<0.001 (AD vs. HC) <0.01 (AD vs. VD) <0.05 (VD vs. HC)	IMR
Yang et al., 2018 [117]	73	AD (21), MCI (29), HC (23)	AD 37.54 (12.29) MCI 32.98 (10.18) HC 18.85 (10.16)	< 0.001 (AD vs. HC + MCI) >0.05 (MCI vs. HC)	IMR
Tsai et al., 2019 [111]	90	AD (37), MCI (40), HC (13)	AD 27.1 (4.8) MCI 24.5 (4.0) HC 22.5 (3.4)	<0.05	IMR
Wang et al., 2014 [85]	273	AD (97), MCI (54), HC (122)	AD 213.95 (44.57) MCI 209.61 (39.65) HC 214.94 (43.23)	0.457 (MCI vs. HC) remaining comparisons *p* > 0.05	ELISA
Krishnan et al., 2014 [87]	105	AD (30), VD (35), HC (40)	AD 458.62 (253.82) VD 718.3 (326.24) HC 879.19 (389.53)	<0.05 (AD vs. VD) <0.001 (AD vs. HC)	ELISA
Jiao et al., 2015 [89]	285	AD (156), HC (129)	AD 227.1 (102.2) HC 181.0 (103.2)	<0.001	ELISA
Shekhar et al., 2016 [118]	113	AD (39), MCI (37), HC (37)	AD 47.49 (9.00) MCI 39.26 (7.78) HC 34.92 (6.58)	<0.001 (AD vs. HC) <0.001 (AD vs. MCI) 0.059 (MCI vs. HC)	ELISA
Rani et al., 2017 [95]	90	AD (45), HC (45)	AD 451.76 (240.82) HC 836.93 (369.31)	<0.001	ELISA
Jiang et al., 2019 [119]	238	AD (110), HC (128)	AD 26.14 (11.52) HC 15.02 (9.04)	<0.001	ELISA
Dage et al., 2016 [120]	439	MCI (161), HC (378)	MCI 4.34 HC 4.14	0.078	Simoa
Mattson et al., 2016 [114]	563 + 547 (two cohorts)	I: AD (179), MCI (195), HC (189), II: AD (61), MCI (212), HC (274)	AD 3.12 (1.50) and 5.37 (2.56) MCI 2.71 (1.32) and 5.46 (2.71) HC 2.58 (1.19) and 5.58 (2.51)	0.0017 and 0.58 (AD vs. MCI + HC)	Simoa
Mielke et al., 2017 [121]	458	MCI (123) HC (335)	MCI 4.5 (1.8) HC 4.2 (1.5)	0.28	Simoa
Deters et al., 2017 [122]	508	AD (168), MCI (174), HC (166)	AD 3.13 (1.3) MCI 2.81 (1.2) HC 2.71 (1)	0.002 (AD vs. MCI + HC)	Simoa
Mielke et al., 2018 [39]	267	AD (40), MCI (57), HC (172)	AD 7.2 (2.8) MCI 5.9 (2.8) HC 5.9 (1.9)	0.029 (AD vs. MCI) 0.001 (AD vs. HC) >0.05 (MCI vs. HC)	Simoa
Foiani et al., 2018 [115]	176	BvFTD (71), PPA (83), HC (22)	BvFTD 1.96 (1.07) PPA 2.65 (2.15) HC 1.67 (0.50)	<0.05 (FTD vs. HC)	Simoa
Shi et al., 2019 [113]	155	MCI (68), HC (87)	MCI 3.71 (2.3) HC 3.56 (1.84)	0.865	Simoa
Kim et al., 2020 [66]	40	AD (20), HC (20)	AD 32.2 (16.4) HC 13.4 (13.2)	<0.001	CNT

SD, standard deviation; CI, confidence interval; AD, Alzheimer’s disease; MCI, mild cognitive impairments; HC, control group.

**Table 4 jcm-09-03452-t004:** T-tau/Aβ42 diagnostic usability studies.

Study	N	Groups (*n*)	T-tau/Aβ42 (SD/CI)	*p*	Method
Krishnan et al., 2014 [87]	105	AD (30), VD (35), HC (40)	AD 3.42 (2.66) VD 5.76 (3.84) HC 15.06 (10.64)	<0.05 (AD vs. VD) <0.001 (AD vs. HC and VD vs. HC)	ELISA
Rani et al., 2017 [95]	90	AD (45), HC (45)	AD 3.08 (2.35) HC 13.36 (4.42)	<0.001	ELISA
Startin et al., 2019 [28]	54	AD (27), HC (27)	AD 10.23 (0.77-52) HC 10.59 (1.14–82.25)	>0.05	Simoa
Kim et al., 2020 [66]	40	AD (20), HC (20)	AD median about 5.5 HC median 2 (read from the figure)	<0.000001	CNT

SD, standard deviation; CI, confidence interval; AD, Alzheimer’s disease; MCI, mild cognitive impairments; HC, control group.

**Table 5 jcm-09-03452-t005:** YKL-40 concentration in serum studies.

Study	N	Groups (*n*)	YKL-40 Concentration in Serum (SD or CI) [ng/mL]	*p*	Method
Craig-Schapiro et al., 2010 [45]	237	AD (CDR1 and CDR0.5), HC	AD (CDR1): 91.9 (15) AD (CDR 0.5): 81.1 (8) HC (CDR 0): 62.5 (3.4)	0.031 (AD CDR1 vs. HC) 0.046 (AD CDR0.5 vs. AD)	ELISA
Choi et al., 2011 [124]	141	AD (61), MCI (41), HC (35)	AD: 376.86 (54.1) MCI:176.49 (25.69) HC: 96.91 (11.02)	0.014 (AD vs. HC) 0.008 (AD vs. MCI)	ELISA
Grewal et al., 2016 [93]	75	15 (aMCI) and 10 (HC) women of white race (CA), Afro-Americans (AA) and people of Latino origin (LA)	LA aMCI 114.08 (30.02) CA aMCI 93.39 (12.70) AA aMCI 54.26 (10.12) LA HC 54.2 (8.37) CA HC 70.92 (15.96) AA HC 54.66 (14.42)	0.033 (LA) 0.418 (CA) 0.988 (AA)	ELISA
Surendranatan et al., 2018 [125]	35	DLB (19), HC (16)	DLB 64.150 (46.616) HC 43.034 (28.357)	0.115	ELISA
Villar-Pique et al., 2019 [126]	315	CJD (78), AD (50), DLB (34), FTD (17), VD (22), ND (44), HC (70)	DLB: 167 (157) CJD: 189 (167) FTD: 125 (108) VD: 140 (150) AD: 133 (110) ND: 95 (61) HC: 84 (84)	<0.001 (CJD vs. HC) remaining *p* > 0.05	ELISA

SD, standard deviation; CI, confidence interval; AD, Alzheimer’s disease; MCI, mild cognitive impairments; HC, control group; CDR, clinical dementia rating; CA, Caucasian race; AA, African American race; LA, Latino-American race; aMCI, amnestic MCI subtype; VD, vascular dementia; CJD, Creutzfeld–Jakob disease; DLB, dementia with Lewy bodies; FTD, frontotemporal dementia; ND, neurological diseases other than dementia.

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
