# Peer review of "Diagnostic Utility of Selected Serum Dementia Biomarkers: Amyloid β-40, Amyloid β-42, Tau Protein, and YKL-40: A Review"

_jcm, 2020, doi:10.3390/jcm9113452_

Round 1

Reviewer 1 Report

The manuscript by Karolina Wilczyńska and Napoleon Waszkiewicz

By analyzing and summarizing cross-sectional studies of dementia published in Medline, PubMed and Web of Science database around 2000-2020, Karolina Wilczyńska and Napoleon Waszkiewicz concluded that t-tau, t-tau/Aβ42 and YKL-40 in serum probably were promising biomarkers for dementia diagnosis, but not for differential diagnosis. Especially, YKL-40 showed more advantages in early MCI diagnostics, while t-tau was more useful in assessment for prodromal states progress of mild AD. 

Overall, the manuscript is well-written and organized. The data presented in the manuscript are of excellent quality.

The major weaknesses are that some of the most prominent studies were not included in the review: 

1)    In line 115-116, the authors stated the popular theory that Aβ is the trigger in AD pathogenesis. However, a recent paper by Daniela  Puzzo, et al. 2020 Aug 10. J Clin Invest, has shown that tau may contribute to AD pathogenesis by a mechanism that is independent of Aβ.

2)    p-tau-217 in blood plasma has been shown to be extremely valuable marker for early AD diagnosis. See Nicolas R. Barthelemy, et al. 2020. J Exp Med. 217 (11): e20200861.

The minor weaknesses are:

1)    In line 43-46, since Huntington’s disease patients also suffer from dementia, I think neurodegenerative dementia should include HD.

2)    In line 252, it is not appropriate to include unpublished data.

3)    In line 325-329, this sentence is too long, hard to read.

4)    In line 372-373, please recount the numbers of the published studies for each category.

5)    In line 445, Should “CSF” be plasma here?

Author Response

Thank you very much for your review. Here are the responses to your comments.

Major weaknesses:

1) „In line 115-116, the authors stated the popular theory that Aβ is the trigger in AD pathogenesis. However, a recent paper by Daniela Puzzo, et al. 2020 Aug 10.J Clin Invest, has shown that tau may contribute to AD pathogenesis by a mechanism that is independent of Aβ”

Added: More and more reports show that the amyloid cascade hypothesis does not fully explain the pathogenesis of AD. It has recently been demonstrated that amyloid-dependent memory and synaptic plasticity impairments can occur without tau.

2) „p-tau-217 in blood plasma has been shown to be extremely valuable marker for early AD diagnosis. See Nicolas R. Barthelemy, et al. 2020. J Exp Med. 217 (11): e20200861.”

Added: „Hyperphosphorylated tau protein and its isoforms such as p-tau-217 measured in the peripheral blood are also promising AD biomarker candidates”

Minor comments:

1) „In line 43-46, since Huntington’s disease patients also suffer from dementia, I think neurodegenerative dementia should include HD” 

HD included

2) „In line 252, it is not appropriate to include unpublished data”

Sentence removed

3) „In line 325-329, this sentence is too long, hard to read”

Divided into several sentences: Review articles, prospective studies were excluded. Studies on biomarkers in other than plasma or serum biological material, as well as studies on specific isoforms of the markers were excluded. We also did not qualify research made on specific groups ie. corpses, children, patients with concomittant diseases (e.g. Down syndrome) or patients with undefined dementia aetiology. Some groups of patients were investigated in more than one article, in such cases only one, methologically best study, was taken into account.

4) „In line 372-373, please recount the numbers of the published studies for each category”

- Non clinically significant – 5 (Wang et al., 2014 [91]; Dage et al., 2016 [115]; Mattson et al., 2016 (cohort II)[117]; Mielke et al., 2017 [118] ; Shi et al., 2019 [112]).

- Tau higher in controls – 2: Krishnan et al., 2014 [93]; Rani et al., 2017 [105]

- Tau lower in controls – remaining 14

Total 21 (one of 20 studies (Mattson et al.,2016 [117]) contained two separate cohorts).

5) „In line 445, Should “CSF” be plasma here?”

Here we refer to amyloid in CSF. We notice that in spite of proven amyloid utility as AD CSF biomarker, the research on its plasma concentration gives incoherent results.

Reviewer 2 Report

Authors complied the many publications on blood biomarkers to suggest a complexity and the usefulness of correlating with other biomarkers. However, authors do not have clear definitions of the different blood media. Authors need to clarify the medium of the blood and discuss their significance. Serum and plasma are very different medium. Also, there are many awkward English and medical expressions.

Author Response

Thank you very much for your review.

As you suggested, English was extensively corrected.

All the reviewed studies (except of the one by Abdullah et al. [71] – which was marked in Table 2) were conducted using serum not plasma and out attention was paid to that. „Plasma” unadequately used in the text was replaced with „serum” (line 203, 240, Table 5).

Reviewer 3 Report

The manuscript entitled “Diagnostic utility of selected dementia biomarkers in serum – a review” is devoted to analyze and to compare the results obtained in different studies on biomarkers used for dementia. The objective is to analyze their validity and application for diagnosis.

An extensive search of the bibliography has been conducted. The interest of the study is high for clinical practice. A final number of 50 studies were included in the analysis devoted to amyloid-beta in their different combinations, 20 studies to t-tau, 5 studies to YKL-40, and 4 studies to the combination of t-tau and amyloid-beta.

In my opinion, the article requires some minor corrections and to include in the analysis the effect of some parameters which have not been included: methodology of the study, age of the groups, and sex. Also, language revision is recommended.

Minor comments are listed below:

TITLE:

Please consider the inclusion of the selected biomarkers in the title for a better description of the study.

ABSTRACT:

  • AD and MCI abbreviations are not fully named at first.

INTRODUCTION

  • 55. The sentence is unclear
  • A Figure or different figures including a diagram of amyloid beta, t-tau and YKL-40 could be helpful for understanding their relevance
  • 188. Please reformulate the sentence. Aging is not a secondary condition to tauopathy…
  • 195. Please revise the denomination of motor diseases
  • 198. In these asymptomatic patients, protein deposition could be assigned to aging?
  • 213. Please substitute thru by though
  • 231. YKL-40
  • 248. Please use the full name of CSF at first
  • 308. BBB: the abbreviation is not fully named at first
  • Table 1 (and general organization of the sections): Why is the combination of t-tau and amyloid beta not placed before YKL-40? It should be more intuitive

RESULTS

  • 347. “39 trials were about AD, 20 - MCI” Please revise the sentence
  • 352. In any cognitive disorder? Please specify
  • With respect to the studies about amyloid markers, AB40, AB42 and AB40/AB42 ratio, the authors, obviously, mention only some of the results found. However, some additional information is needed. For example, in 10/29 studies about the ratio, the results were not statistically significant, in 12/29, there were statistical differences, being the “AB42/AB40” higher than in control groups and in 6/29 test groups. What happens in the remaining study?

    Also, please revise the ratio: it is written AB42/AB40 and not AB40/AB40. Is it a mistake writing the ratio or it corresponds to the ratio analyzed? Since it could lead to opposite results…

  • All Tables in the Results section: Papers are listed in chronological order. However, it could be more visual if the authors listed them firstly by the method of analysis and then by date in order to better visualize if there are differences due to the techniques.

  • 370. Simoa: use the non-abbreviated in the first mention

  • In all biomarkers analyzed, the author should include in the description of the statistical differences between groups the method used. I mean, are the differences in results found attributable to differences in the methodology?

  •  383 and 396. I believe that AB42 ratio should be t-tau/AB42 ratio

  • Table 4. Please revise the data from Kim et al., Also, revise non-English comments and abbreviations.

  • YKL-40. In order to follow a similar scheme to other sections, please avoid numbering the papers.

  • 414. “suggests that”

DISCUSSION

  • 445. suggest that
  • 446. probable that
  • 449. “Typical for”
  • In general, the discussion is very brief. The authors should analyze and comment more extensively on their results. Again, is there any effect on the methodology? Also, is there any effect of age, sex or type of dementia?

Author Response

Thank you very much for your review. Here are the responses to your comments.

TITLE:

Please consider the inclusion of the selected biomarkers in the title for a better description of the study.”

Included

ABSTRACT:

AD and MCI abbreviations are not fully named at first.

Full names added

INTRODUCTION

55. The sentence is unclear

Rewritten „Contrary to Wernicke-Korsakoff syndrome associated primarily with memory impairment, in alcoholic dementia there are disorders in many other cognitive functions.”

A Figure or different figures including a diagram of amyloid beta, t-tau and YKL-40 could be helpful for understanding their relevance

A figure added

188. Please reformulate the sentence. Aging is not a secondary condition to tauopathy…

„physiological aging” removed from the sentence

195. Please revise the denomination of motor diseases
Amyotrophic lateral sclerosis”, „parkinsonism-dementia complex”, „postencephalitic parkinsonism”, „diffuse neurofibrillary tangles with calcification”, „subacute sclerosing panencephalitis and „non -guanamian motor neuron disease with neurofibrillary tangles” were revised.

198. In these asymptomatic patients, protein deposition could be assigned to aging?
We think not, because according to the cited paper (Fernando et al., 2004 [7] Table 4 line 1) the severity of neurofibrillary lesions was not correlated with the age at death.

213. Please substitute thru by though

done

231. YKL-40

done

248. Please use the full name of CSF at first
done

308. BBB: the abbreviation is not fully named at first
corrected

Table 1 (and general organization of the sections): Why is the combination of t-tau and amyloid beta not placed before YKL-40? It should be more intuitive We decided to place all the „single” markers before the combined markers. Anyway, we would not mind if the editor deems it appropriate to change the order of sections and tables.

RESULTS

347. “39 trials were about AD, 20 - MCI” Please revise the sentence
Transformed into: „In 39 trials AD patients were investigated. 20 trials compared MCI subjects to cognitively normal controls.”

352. In any cognitive disorder? Please specify
Transformed into :dementia or MCI”

With respect to the studies about amyloid markers, AB40, AB42 and AB40/AB42 ratio, the authors, obviously, mention only some of the results found. However, some additional information is needed. For example, in 10/29 studies about the ratio, the results were not statistically significant, in 12/29, there were statistical differences, being the “AB42/AB40” higher than in control groups and in 6/29 test groups. What happens in the remaining study?

The Aβ42/Aβ40 studies summary was revised: „Amyloid peptides concentration ratio was assessed in 29 studies. In 9 cases the results were not statistically significant. Among the comparisons that were statistically significant, in 11 cases Aβ42/Aβ40 concentration ratio was higher in control groups and in 7 cases – in study groups. In two of the studies the Aβ42/Aβ40 ratio was similar in AD and control groups and significantly higher than in MCI groups [68, 103].”

Also, please revise the ratio: it is written AB42/AB40 and not AB40/AB40. Is it a mistake writing the ratio or it corresponds to the ratio analyzed? Since it could lead to opposite results…

In the minority of studies the ratio was calculated as AB40/42, which was marked in the table. We decided not to transform the values given in original papers, but we took into account they had to be interpreted as opposite, i.e. if AB40/42 was higher in AD then AB42/40 was higher in controls.

All Tables in the Results section: Papers are listed in chronological order. However, it could be more visual if the authors listed them firstly by the method of analysis and then by date in order to better visualize if there are differences due to the techniques.
Changed as suggested.

370. Simoa: use the non-abbreviated in the first mention
Added in line 342

In all biomarkers analyzed, the author should include in the description of the statistical differences between groups the method used. I mean, are the differences in results found attributable to differences in the methodology?
Added.
AB40:
The inconsistence of results did not depend of the method used. A large part of research gave statistically not significant results (12 out of 20 for ELISA, 3 out of 8 for IMR, 2 out of 5 for xMAP).
AB42:Among methods repeated in several studies, it draws attention that in all seven studies conducted with the use of IMR method Aβ42 levels were significantly higher in AD or MCI groups than in controls. On the other hand, studies based on ELISA or xMAP method gave very inconsistent results.”
AB42/40: „The results varied depending on laboratory methods use. None of the 3 studies using xMAP gave statistically significant results, while in 4 out of 5 IMR studies Aβ42/Aβ40 ratio was significantly higher in AD or MCI than in controls. The results obtained in ELISA studies were highly inconsistent.”

Tau: „In all 6 studies using IMR method and 5 out of 6 performed with ELISA determination, tau levels were significantly higher in AD/MCI patients. Simoa gave less consistent results – in 3 out of 8 cohorts the differences between groups did not reach the statistical significance.”

383 and 396. I believe that AB42 ratio should be t-tau/AB42 ratio
Corrected

Table 4. Please revise the data from Kim et al., Also, revise non-English comments and abbreviations.

Corrected

YKL-40. In order to follow a similar scheme to other sections, please avoid numbering the papers.
Numeration removed

414. “suggests that”

Corrected

DISCUSSION

445. suggest that

corrected

446. probable that

corrected

449. “Typical for”

corrected

In general, the discussion is very brief. The authors should analyze and comment more extensively on their results. Again, is there any effect on the methodology? Also, is there any effect of age, sex or type of dementia?

Added: „Considering the results of amyloid markers separately for individual laboratory methods, we noticed some differences. In the case of amyloid markers, almost all tests performed with the IMR method gave consistent, statistically significant results, in contrast to the determinations by ELISA or xMAP. On the other hand, the Simoa method, often used for the determination of tau protein concentrations, in the analyzed works gave much less consistent results compared to ELISA and IMR. Therefore, it seems that the laboratory method used may influence the obtained results.”

The effects of dementia type is hard to analyze. Only a few single of cited works included patients with other than AD dementias (eg. VD, FTD). Those data seem to be insufficient to pose reasonable conclusions.